# Transformer Working Memory Enables Regular Language Reasoning And Natural Language Length Extrapolation

**Ta-Chung Chi**
Carnegie Mellon University
`tachungc@andrew.cmu.edu`

**Ting-Han Fan**
Princeton University
`tinghanf@princeton.edu`

**Alexander I. Rudnicky**
Carnegie Mellon University
`air@cs.cmu.edu`

**Peter J. Ramadge**
Princeton University
`ramadge@princeton.edu`

## Abstract

Unlike recurrent models, conventional wisdom has it that Transformers cannot perfectly model regular languages. Inspired by the notion of working memory, we propose a new Transformer variant named RegularGPT. With its novel combination of Weight-Sharing, Adaptive-Depth, and Sliding-Dilated-Attention, RegularGPT constructs working memory along the depth dimension, thereby enabling efficient and successful modeling of regular languages such as PARITY. We further test RegularGPT on the task of natural language length extrapolation and surprisingly find that it rediscovers the local windowed attention effect deemed necessary in prior work for length extrapolation.

## 1 Introduction

It is long believed that *Working Memory* (WM), a term coined in 1960s to liken human minds to computers, plays an important role in humans' reasoning ability and the guidance of decision-making behavior (Baddeley and Hitch, 1974; Baddeley, 1992; Ericsson and Kintsch, 1995; Cowan, 1998; Miyake et al., 1999; Oberauer, 2002; Diamond, 2013; Adams et al., 2018). While no single definition encompasses all applications of WM (Adams et al., 2018), the following one should be shared by all of the theories of interest:

> *Working memory is a system of components that holds a **limited amount** of information temporarily in a heightened state of availability for use in ongoing processing.* - Adams et al. (2018)

WM is instantiated in the two major driving forces of sequence modeling: Recurrent neural networks'(RNN) (Elman, 1990; Jordan, 1997; Hochreiter and Schmidhuber, 1997) short term memory modulated by their recurrent nature and gate design (Rae and Razavi, 2020a; Nematzadeh

et al., 2020; Armeni et al., 2022), and Transformers' (Vaswani et al., 2017) salient tokens heightened by self-attention.

In reality, self-attention often attends broadly (Clark et al., 2019), violating the limited amount of information notion of WM. Our hypothesis is that such violation is to blame for Transformers' failure on algorithmic reasoning of regular languages (Deletang et al., 2023; Liu et al., 2023) such as PARITY, a seemingly simple task that checks if the number of 1s in a bit string is even. Surprisingly, a Transformer can only count the number of 1s correctly when the sequence length is held fixed at training sequence length $T_{tr}$, and it fails miserably when the testing sequence length extrapolates to $T_{ex} > T_{tr}$ (Hahn, 2020; Bhattamishra et al., 2020; Chiang and Cholak, 2022; Deletang et al., 2023; Liu et al., 2023). In contrast, an RNN can extrapolate perfectly.

The goal of this work is therefore to improve Transformers' WM by limiting the amount of accessible information at a time. Existing attempts that use a combination of scratchpad and recency biases (Wei et al., 2022; Nye et al., 2022; Anil et al., 2022; Liu et al., 2023) is not optimal as it completely foregoes the parallelization property of a Transformer, making it as computationally inefficient as an RNN.

This begs the question: *Does there exist a more efficient Transformer working memory design?* The answer is affirmative thanks to the proposed **RegularGPT**, which boils down to the three design choices: Weight-Sharing, Adaptive-Depth, and Sliding-Dilated-Attention; Each of them has been proposed previously but it is the unique combination that sparks the successful and efficient learning of regular languages. We will further demonstrate its: 1) similar recursive parallel structure as linear RNN (Orvieto et al., 2023), resulting in $\log T_{tr}$ or $\log T_{ex}$ layers, and 2) generalizability by showing strong performance on the task of Transformer

natural language length extrapolation (Press et al., 2022; Chi et al., 2022, 2023).

In this work, we use $[N]$ to denote the list of non-negative integers $[0, \ldots, N-1]$. The Transformer model used in this work is always causal. It takes in an input sequence of $T \leq T_{tr}$ units (can be tokens or bits) $\sigma_{i \in [T]}$, passes them through a fixed amount of $L$ transformer layers, and finally computes the distribution over the vocabulary $V$ via the prediction head $W_o$.

## 2 Background

### 2.1 Regular Language and Algorithmic Reasoning

The Chomsky hierarchy (Chomsky, 1956b) classifies formal languages into different hierarchies based on their increasing complexity. Each hierarchy represents a family of formal languages that can be solved by the corresponding automaton. At the lowest level resides the family of regular languages, which can be expressed using a finite state automaton (FSA), a computational model comprising a set of states and transitions connecting them.

Our primary objective is to enhance the algorithmic reasoning of the Transformer model on regular languages by testing its language transduction capability under the extrapolation setting. Concretely, the model is trained only to predict desired outputs on a set of short length-$T$ sequences with $T \leq T_{tr}$. Still, it must also predict the correct outputs for longer testing sequences of length $T_{ex} \gg T_{tr}$. It is worth noting that we evaluate our model via language transduction following recent work (Deletang et al., 2023; Liu et al., 2023), instead of the conventional language recognition protocol. Both settings are equally hard as they are underpinned by the same finite state semiautomaton. Interested readers may refer to Deletang et al. (2023) for further details regarding the two evaluation protocols. We also reveal the connection between RegularGPT and finite state semiautomaton later in §7.

### 2.2 Failure Mode and An Inefficient Fix

The PARITY task involves a length $T$ bit string $\sigma_1 \sigma_2 \cdots \sigma_T$ where each bit $\sigma_i$ is randomly sampled from a Bernoulli distribution with $\mathbb{P}(\sigma_i = 1) = 0.5$. The goal is to determine whether the sequence contains an even or odd number of 1s.

It has been observed that a Transformer is incapable of performing length extrapolation on PARITY, but what could be its potential failure mode?

Previous work sheds light on this by showing that a Transformer might settle on the naive-summation approach (Anil et al., 2022; Deletang et al., 2023; Liu et al., 2023). Concretely, it sums up all the bits and outputs the summation modulo 2. This approach fails since unseen summations will be produced when the model takes sequences of length $T_{ex} > T$ as input or $\mathbb{P}(S_i)$ deviates from 0.5.

To the best of our knowledge, the existing remedy (Liu et al., 2023; Anil et al., 2022) is to use scratchpad (Wei et al., 2022; Nye et al., 2022) along with recency biases (Press et al., 2022) to enforce the correct learning: They create a scratchpad that interleaves the sequence of input bits and intermediate answers $(\sigma_1, q_1, \sigma_2, q_2, \cdots, \sigma_T, q_T)$, where $q_i = \text{solve}(\sigma_1 \cdots \sigma_i)$. The model is trained to predict all the $\sigma_{i \in [T]}$. Recency biases play the role of limiting a Transformer's receptive field to only a few most recent $\sigma$ and $q$ at every timestep $i$. This is to prevent self-attention from ignoring $q$ and giving the same naive-summation solution.

Scratchpad and recency biases jointly create the notion of WM along the temporal dimension similar to RNNs, thereby enabling successful extrapolation on regular languages. Nevertheless, we note that this fix is inefficient during inference since all the intermediate answers $q_i$ have to be generated sequentially before reaching the final answer $q_T$. A desirable fix should only take in the input bits $(\sigma_1, \sigma_2, \cdots, \sigma_n)$ and directly generate the final answer $q_T$. In other words, our goal is to find an efficient WM design for a Transformer.

### 2.3 A Desirable Fix for PARITY (Figure 1)

An alternative solution to the PARITY problem is based on the spirit of divide-and-conquer, where we first divide the sequence into $T/C$ chunks with each chunk of length $C < T$, and we compose the final answer by recursively merging the chunk outputs. This approach does not suffer from the unseen summation issue as the model was trained to handle a fixed amount of $C$ bits at a time in its WM (chunk). It then recursively applies the already-seen results to compose the final solution when it encounters longer sequences during inference. More importantly, it is more efficient than the scratchpad and recency biases approach since it only requires $\log_C T$ layers of parallel computations instead of $2T$ steps of sequential decoding.

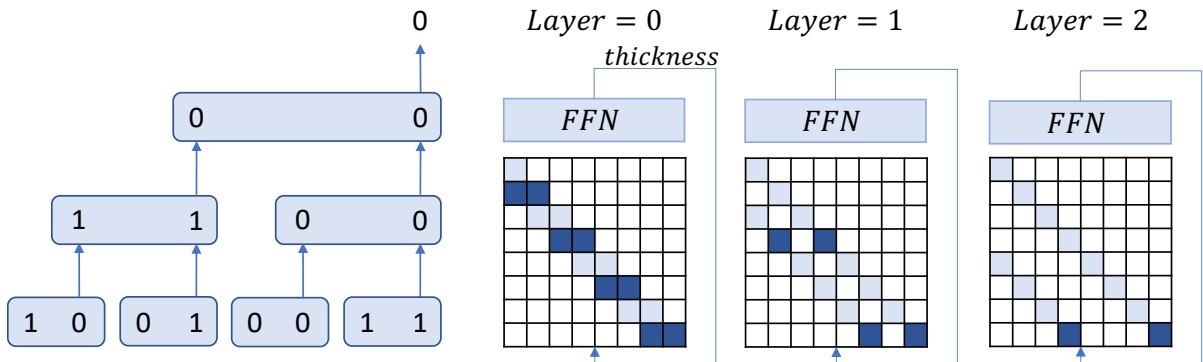

Figure 1: This is the divide-and-conquer approach that solves the PARITY problem. The lightly shaded blue cells represent $M_{mn}^{(l)}$ in eq. (1). The darkened blue cells represent the routing path to solve the result for the last bit specifically. As we can see, this approach requires at most $\log_2 T$ layers to obtain the result for a length $T$ input sequence, rendering it a more efficient approach compared to the combination of scratchpad and recency biases.

## 3 Proposed Architecture of RegularGPT

We present our modifications to the vanilla Transformer below. Only the related operations will be expanded, and we follow all the other details of GPT2 (Radford et al., 2019).

### 3.1 Sliding-Dilated-Attention

A Transformer layer at layer $l$ consists of a self-attention operation denoted as $\text{SA}^{(l)}$ and feed-forward network denoted as $\text{FFN}^{(l)}$. Originally, $\text{SA}^{(l)}$ computes the inter-token relationships across all $T$ units. Instead, we set the chunk size to $C$ and produce $T/C$ non-overlapping chunks;[1] Only the units within the same chunk inter-attend with each other. In practice, this can be achieved by an attention mask $M^{(l)} \in \mathbb{R}^{T \times T}$ at layer $l$. $M^{(l)}$ shares the same shape as the self-attention matrix (see Figure 1) and is defined as:

$$M_{mn}^{(l)} = \begin{cases} r_{(m-n)/C^\ell}, & \text{if } \frac{m-n}{C^l} \in [C] \\ -\inf, & \text{otherwise} \end{cases} \quad (1)$$

Note that $M$ is a lower triangular matrix due to the causal nature of our model. $r_i$'s with $i \in [C]$ are learnable relative positional scalars. To be precise, each attention head has a different set of learnable biases $r_i$'s. Here, we drop the dependency on the head for notational simplicity.

The use of $r_i$'s is similar to the positional scalars of T5 (Rae and Razavi, 2020a) except that we do not use the log-binning strategy over $m - n$. It is to facilitate the extraction of global information instead of enforcing the windowed-attention effect (Raffel et al., 2020; Press et al., 2022; Chi

---

[1]Whenever $T$ is not divisible by $C$, we pad the input sequence such that its length is a multiple of $C$.

et al., 2022, 2023). $M$ will then be added to the original self-attention matrix, creating the proposed Sliding-Dilated-Attention effect. The output of $\text{SA}^{(l)}$ will be transformed by the positional-independent $\text{FFN}^{(l)}$ to produce $o_{i \in [T]}^{(l)}$.

The case of $C = 2$ is used as a possible construction of Theorem 1 in Liu et al. (2023). However, their focus is not on length extrapolation, hence lacking the below two proposed modifications.

### 3.2 Adaptive-Depth and Weight-Sharing

Since our Sliding-Dilated-Attention limits the number of accessible tokens at a time, we need an adaptive depth $\bar{L} = \log_C T$ so that the final output can utilize every single piece of input information. However, when $T_{ex} > T_{tr}$, the depth during inference will be higher than that during training. The simplest way to solve this challenge without further parameter updating is to perform Weight-Sharing across layers. To account for the possible performance loss due to Weight-Sharing, we first thicken the model by $K$ times, resulting in a total number of $K \cdot \bar{L}$ layers. Next, we share the weights across the $K \cdot \bar{L}$ layers in the following way for $k \in [K]$:

$$\text{SA}^{(l \cdot K + k)} = \text{SA}^{(k)} \quad \text{for } l \in [\bar{L}]$$
$$\text{FFN}^{(l \cdot K + k)} = \text{FFN}^{(k)} \quad \text{for } l \in [\bar{L}]$$

It can be equivalently interpreted as stacking more SA and FFN components within every Transformer layer, and the same thickened layer is reused $\bar{L}$ times. This layer thickening design is only used in the natural language modeling experiments in §6.

### 3.3 Where is the WM Notion?

Instead of instantiating WM along the temporal dimension as the combination of scratchpad and

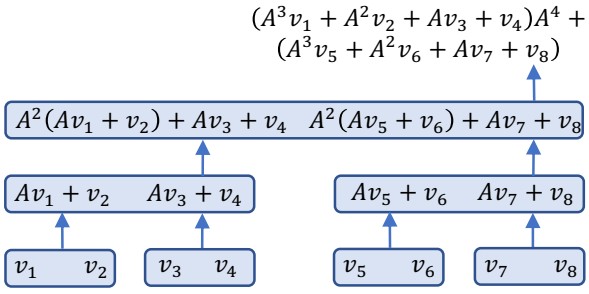

Figure 2: This is the parallel scan algorithm that can accelerate a linear RNN. In this example, we visualize the routing path for computing $x_8$. Blocks at the same layer can be computed in parallel on GPUs.

recency biases, RegularGPT limits the amount of information along the depth dimension. As we have seen, the idea of breaking $T$ units into several chunks limits the amount of accessible information at each layer, thereby enabling the WM notion. A similar argument was made by Yogatama et al. (2021) in a sense that they categorized Longformer (Beltagy et al., 2020), a transformer variant with local attention pattern, as a model of working memory. Finally, thanks to modern accelerators such as GPU, all chunks at a layer can be processed concurrently, and this further makes RegularGPT more favorable over the scratchpad and recency biases approach.

## 3.4 Complexity Analysis

The sparse attention pattern of RegularGPT suggests it might enjoy the same speedup provided by sparsified Transformers. The complexity of our model is $O(TCK \log_C T)$ where $TC$ is the complexity of each self-attention module and $K \log_C T$ is the total number of layers. On the other hand, the vanilla Transformer follows $O(T^2 L)$. To illustrate the possible speedup, if $T = 512$ and $C = 128$, then $512 \cdot 128 \cdot K \log_{128} 512 < 512^2 L$ when $K < \frac{512L}{128 \log_{128} 512} \approx 3.11L$. Namely, as long as $K < 3L$, our model is likely to be more efficient than a vanilla Transformer.

## 4 Connection to Prior Work

**Sliding-Dilated-Attention** This special attention pattern dates back to pre-Transformer era such as Wavenet (van den Oord et al., 2016) with dilated convolution. It can also be viewed as a special form of Longformer attention pattern with systematic dilation (Beltagy et al., 2020).[2] Limiting the

range of attention in lower layers of a Transformer is also corroborated in Rae and Razavi (2020b), where they find such design does not deteriorate the performance.

**Adaptive-Depth and Weight-Sharing** AL-BERT (Lan et al., 2020) and Universal Transformer (Dehghani et al., 2019) share the parameters across layers. The weight sharing design makes them compatible with the idea of Adaptive Computation Time (Graves et al., 2014) and Dynamic Halting (Dehghani et al., 2019; Elbayad et al., 2020), which allocate different computational budget depending on the complexity of tasks (Simoulin and Crabbé, 2021; Csordás et al., 2022). However, they lack the special Sliding-Dilated-Attention design that is necessary for ruling out naive solutions.

**Linear RNN** Given $x_0 = 0 \in \mathbb{R}^N$ and the input vectors $u_1 \cdots u_T$, a linear RNN (Orvieto et al., 2023) for $k \in [T]$ can be written as:

$$x_k = Ax_{k-1} + Bu_k = \sum_{j=0}^{k-1} A^j Bu_{k-j}$$

$$= \sum_{j=0}^{k-1} A^j v_{k-j},$$

where we set $v_{k-j} = Bu_{k-j}$. The operation can be accelerated by the parallel scan algorithm that permits efficient cumulative sum (Ladner and Fischer, 1980; Blelloch, 1990; Lakshmivarahan and Dhall, 1994; Martin and Cundy, 2018; Liu et al., 2023; Smith et al., 2023). As we can see in Figure 2, the routing path specified by the parallel scan algorithm is the same as our Sliding-Dilated-Attention illustrated in Figure 1.

## 5 Regular Language Experiments

### 5.1 Task Descriptions

We focus on the four tasks in section 1) (Deletang et al., 2023) of Table 1 as they will also be used in our analysis in §7. For tasks in section 2), please refer to Bhattamishra et al. (2020) for details.

**Even Pairs** A model needs to predict whether the total count of "ab" and "ba" pairs is even. In the example of "aabba", there is one "ab" and one "ba", resulting in a total count of 2, which is even. This task is equivalent to checking whether the first and last characters in a string are identical.

---

[2]The original Longformer also adopts dilated attention on a few heads at higher layers but without the systematic pattern

used in this work.

| Task | RNN | Transformer | RegularGPT $C = 2$ | $C = 3$ |
|---|---|---|---|---|
| *1) Deletang et al.* | | | | |
| Even Pairs | 100.0 / 100.0 | 99.7 / 73.2 | 100.0 / 89.3 | 100.0 / 96.6 |
| Modular Arithmetic | 100.0 / 100.0 | 21.9 / 20.3 | 96.4 / 82.6 | 21.2 / 20.5 |
| Parity Check | 100.0 / 98.9 | 52.3 / 50.1 | 100.0 / 100.0 | 100.0 / 88.7 |
| Cycle Navigation | 100.0 / 100.0 | 21.7 / 20.6 | 100.0 / 100.0 | 100.0 / 78.6 |
| *2) Bhattamishra et al.* | | | | |
| $D_2$ | 100.0 / 100.0 | 100.0 / 80.1 | 100.0 / 100.0 | 99.8 / 96.5 |
| $D_3$ | 100.0 / 100.0 | 100.0 / 77.8 | 100.0 / 99.7 | 98.6 / 93.0 |
| $D_4$ | 100.0 / 100.0 | 100.0 / 82.6 | 100.0 / 98.7 | 97.7 / 91.6 |
| $D_{12}$ | 100.0 / 100.0 | 100.0 / 80.3 | 100.0 / 99.8 | 94.1 / 90.4 |
| Tomita 3 | 100.0 / 100.0 | 100.0 / 94.4 | 100.0 / 99.7 | 100.0 / 99.9 |
| Tomita 4 | 100.0 / 100.0 | 100.0 / 70.0 | 100.0 / 99.8 | 100.0 / 99.3 |
| Tomita 5 | 100.0 / 100.0 | 74.5 / 74.5 | 100.0 / 99.8 | 98.2 / 84.1 |
| Tomita 6 | 100.0 / 100.0 | 50.0 / 50.0 | 100.0 / 98.5 | 100.0 / 65.7 |

Table 1: **Length generalization results on Regular Languages (Max/Avg).** All models in the first section (Deletang et al.) are trained on sequences of length 40. The reported numbers are the average of length extrapolation results from 41 to 500. Each result is an average over 3 seeds. All models in the second section (Bhattamishra et al.) are trained on sequences of length 50. The reported numbers are the average of length extrapolation results from 51 to 100. Each result is an average over 3 seeds. Please refer to Appendix A for the detailed hyperparameters.

**Modular Arithmetic** Given a sequence of numbers in {0, 1, 2, 3, 4} and operations in {+, -, ·}, a model needs to compute the result modulo 5. For example, $x = 1 + 2 - 4$ evaluates to $y = 4$.

**Parity Check** A model needs to compute whether the number of bs in a given binary string is even. For example, the sequence x = aaabba contains 2 bs, which is even.

**Cycle Navigation** Given a sequence of movements on a cycle of length 5, a model needs to compute the end position. The possible movements are STAY, INCREASE, DECREASE encoded as {0, 1, 2}. The agent always starts at position 0. For example, 010211 means the agent stops at position $2 = 0 + 1 + 0 - 1 + 1 + 1$.

### 5.2 Language Transduction and Extrapolation

First, we want to know if endowing a Transformer with the notion of WM really improves its length extrapolation capability on regular languages. We test RegularGPT and all the baselines on two sets of regular languages from prior work (Deletang et al., 2023; Bhattamishra et al., 2020).[3] Prior work often

reports the maximum score across different hyperparameter settings and random seeds because their goal is to know if a model can extrapolate *at all*. We additionally report the average scores since we want to know if the model can consistently obtain good performance. The baseline models we compare against are an RNN and vanilla Transformer with Transformer-XL style relative positional embedding (Dai et al., 2019). Table 1 shows that RegularGPT with $C = 2$ acheives similar performance as an RNN and substantially outperforms a vanilla Transformer.

### 5.3 The Effect of Chunk Size $C$

We vary the chunk size $C$ of RegularGPT to see its impact on the performance. The motivation for using a larger $C$ is to reduce the number of layers (i.e., $\bar{L} = \log_C T$ decreases in $C$) and increase the degree of parallelization. However, in Table 1, a larger $C$ seems to pose a challenge to RegularGPT on the Modular Arithmetic task. Modular Arithmetic is a hard task with far more states and complicated state transitions. Increasing $C$ is likely to increase the task difficulty by composing more state transitions at once. We will have an in-depth discussion of the theoretical reasons in §7.

---

[3]Our implementation is based on the codebase of Deletang et al. (2023) at: `https://github.com/deepmind/neural_networks_chomsky_hierarchy`. We additionally implement the regular languages in the second section

of Table 1.

| Settings | Probability $\mathbb{P}(\sigma_i = 1)$ | | | | |
|---|---|---|---|---|---|
| | 0.1 | 0.3 | 0.5 | 0.7 | 0.9 |
| *1) Same Length* | | | | | |
| RegularGPT | 100 | 100 | 100 | 100 | 100 |
| RNN | 100 | 100 | 100 | 100 | 100 |
| Transformer | 98.4 | 99.8 | 99.6 | 97.8 | 77.2 |
| *2) Extrapolation* | | | | | |
| RegularGPT | 100 | 100 | 100 | 100 | 100 |
| RNN | 100 | 100 | 100 | 100 | 100 |
| Transformer | 50.1 | 49.7 | 50.3 | 49.9 | 50.0 |

Table 2: We alter the probability $\mathbb{P}(\sigma_i = 1)$ used to sample 1s of PARITY. The same length setting is 40. The extrapolation setting is from 41 to 500. Each entry is an average over 3 seeds.

## 5.4 Robust to Probability Changes

Other than the length extrapolation experiment, we alter the probability of sampling 1s of PARITY, i.e., set $\mathbb{P}(\sigma_i) \neq 0.5$. The results in Table 2 show that RegularGPT is robust to different sampling probabilities, indicating its successful modeling of the underlying regular language grammar. In contrast, a vanilla Transformer model struggles to achieve good performance even for the same length setting, again validating the fact that it only finds the naive-summation solution as discussed in §2.2.

## 6 Natural Language Experiments

Given that RegularGPT has been battle-tested on the main experiment of regular languages, we now shift gear to benchmark its performance in the natural language scenario. Given a model trained on sequences of length $T_{tr}$, we test it on much longer sequences of length $T_{ex} \gg T_{tr}$ during inference, and the goal is to observe similar perplexities. To optimize efficiency, we employ a random selection process to extract 1,000 chunks, each with $T_{ex}$ tokens from the testing set. Subsequently, we calculate the average perplexity of the last tokens within these chunks to ensure each of them has $T_{ex} - 1$ tokens as the context, thereby avoiding the issue of early token curse (Press et al., 2022; Chi et al., 2023). We compare our model against the existing methods that are known to demonstrate the ability of length extrapolation including T5 (Raffel et al., 2020), ALiBi (Press et al., 2022), and KERPLE (Chi et al., 2022).[4] To counteract the loss of

---

[4] We use the nanoGPT codebase: https://github.com/karpathy/nanoGPT, and the Open-WebText2 dataset: https://huggingface.co/datasets/the_pile_openwebtext2.

expressive power due to weight sharing, we thicken each layer of RegularGPT to $K$ as detailed in §3.

In Table 3, we first observe exploding perplexities for $C = 32$ after $T_{ex} \geq 2048$. RegularGPT might only learn to model $\lceil \log_{32} 512 \rceil = 2$ layers during training, hence it fails to recursively model more than $32^2 = 1024$ tokens during inference. This is validated by $C = 64$ since this time it is able to extrapolate until $64^{\lceil \log_{64} 512 \rceil} = 4096$. While the above argument seems to suggest large $C$, setting $C = 256$ also deteriorates the performance. This might be due to the limited number of chunks ($512/256 = 2$) and $r_i$'s (in Eq. (1)) observed at the second layer, making the learning of $r_i$'s harder. Overall, $C$ is a hyperparameter that needs to be carefully decided for RegularGPT on natural languages. We also observe that 128/12 performs better than 128/6, implying RegularGPT's performance could be improved by stacking more layers to counteract the performance loss due to Weight-Sharing.

It is worth noting that 128/12 performs relatively well and is close to previous methods designed specifically for the task of natural language extrapolation. We will analyze its inner workings in depth in Figure 4 and §7, in which we find that RegularGPT learns the similar local receptive field as prior work, which is likely the key to its successful natural language extrapolation performance.

## 7 Discussion and Analysis

### 7.1 Regular Language and Finite State Semiautomaton

Regular language is the type of formal language recognized by an FSA (Chomsky, 1956a), which is a 5-tuple $(Q, \Sigma, \delta, q_0, F)$, where $Q$ is a finite non-empty set of states, $\Sigma$ is a finite non-empty set of symbols, $q_0 \in Q$ is an initial state, $\delta : Q \times \Sigma \to Q$ is a transition function; $F \subseteq Q$ is a set of final states. However, some of our tasks are better modeled by a finite-state transducer (FST) as discussed in §2.1. To underpin both FSA and FST, we consider a semiautomation $\mathcal{A} = (Q, \Sigma, \delta)$ (i.e., an FSA without $q_0$ and $F$) and establish its connection to a Transformer model.

Let $\sigma_{a:b}$ be the sequence from position $a$ (inclusive) to $b$ (exclusive) out of a length $T$ input sequence (i.e., $0 \leq a < b \leq T$). We define $\mathcal{A}(\sigma_{a:b}) : Q \to Q$ as the $(b-a)$-step state transition relation after receiving $\sigma_{a:b}$.

$$\mathcal{A}(\sigma_{a:b}) = \delta(\cdot|\sigma_{b-1}) \circ \cdots \circ \delta(\cdot|\sigma_a),$$

| $T_{ex}$ | KERPLE | T5 | ALiBi | RegularGPT ($C/K$) | | | | |
|---|---|---|---|---|---|---|---|---|
| | | | | 32 / 6 | 64 / 6 | 128 / 6 | 128 / 12 | 256 / 6 |
| 512 | 24.71 | 24.50 | 24.53 | 32.06 | 30.17 | 28.80 | 26.37 | 27.90 |
| 1024 | 24.42 | 24.38 | 24.90 | 32.03 | 30.30 | 28.94 | 26.91 | 34.38 |
| 2048 | 24.21 | 25.01 | 25.08 | 791.74 | 30.56 | 29.14 | 27.08 | 34.85 |
| 4096 | 24.53 | 28.91 | 25.08 | 812.00 | 30.80 | 29.25 | 27.28 | 35.11 |
| 8192 | 24.74 | 39.08 | 25.08 | 818.49 | 1175.91 | 29.41 | 27.39 | 35.42 |

Table 3: **Natural language extrapolation results on OpenWebText2.** The training length is 512. The numbers are averaged over three random seeds. Please refer to Appendix B for the detailed hyperparameters.

where $f(\cdot) \circ g(\cdot) = f(g(\cdot))$ denotes function composition. With abuse of notation, we define $\mathcal{A}_q(\sigma_{a:b}) \in Q$ as the state after receiving $\sigma_{a:b}$ if starting at $q \in Q$

$$\mathcal{A}_q(\sigma_{a:b}) = \delta(\cdot|\sigma_{b-1}) \circ \cdots \circ \delta(\cdot|\sigma_a), q_0 = q.$$

### 7.2 Modeling Transition Composition

We want to show that the layers of RegularGPT with chunk size $C = 2$ can model the composition of two transition functions:

$$\mathcal{A}(\sigma_{a:b}) = \mathcal{A}(\sigma_{i:b}) \circ \mathcal{A}(\sigma_{a:i}) \text{ for } i \in [a+1, \ldots, b).$$

This way, the regular language problem can be solved recursively using the construction outlined in §3 and Figure 1. To formalize the statement, we first observe that $\mathcal{A}(\sigma_{a:b})$, $\mathcal{A}(\sigma_{a:i})$, and $\mathcal{A}(\sigma_{i:b})$ can be represented in $\mathbb{R}^{|Q|^2}$:

$$\mathcal{A}(\sigma_{a:b}) = \begin{bmatrix} \text{OneHot}_{|Q|}(\mathcal{A}_{q_0}(\sigma_{a:b})) \\ \text{OneHot}_{|Q|}(\mathcal{A}_{q_1}(\sigma_{a:b})) \\ \ldots \\ \text{OneHot}_{|Q|}(\mathcal{A}_{q_{|Q|-1}}(\sigma_{a:b})) \end{bmatrix} \in \mathbb{R}^{|Q|^2},$$
(2)

where $\text{OneHot}_{|Q|}(i)$ is a one-hot vector of length $|Q|$ with the $i$-th index being 1.

The next step is to mix $\mathcal{A}(\sigma_{a:i})$ and $\mathcal{A}(\sigma_{i:b})$ together and get $\mathcal{A}(\sigma_{a:b})$. We show in Lemma 1 that a 2-layer ReLU network can learn (and so can a transformer layer) the composition. The proof of Lemma 1 is deferred to Appendix C.

**Lemma 1** (Approximation for Binary Matrix Product). *Let* $A, B \in \{0,1\}^{n \times n}$ *be binary matrices of dimension* $n \times n$. *Then, there exists a two-layer ReLU network such that*

$$f_{mlp}([Flat(A), Flat(B)]) = Flat(AB),$$

*where* $Flat(X)_{(i-1)n+j} = X_{i,j}$ *for* $i, j \in [n]$ *is the operation that flattens a matrix into a vector.*

Now, we can relate Lemma 1 to the FFN layers in RegularGPT. Following §3, when chuck size $C = 2$ and thickness $K = 1$, the output vector $o_i^{(l)}$ depends on input sequence $\sigma_{i-2^{l+1}+1:i+1}$. Also, $o_i^{(l)}$ is computed from $o_{i-2^l}^{(l-1)}$ and $o_i^{(l-1)}$, which depend on input sequences $\sigma_{i-2^{l+1}+1:i-2^l+1}$ and $\sigma_{i-2^l+1:i+1}$, respectively. This observation implies that $o_i^{(l)}$ likely models the transition function $\mathcal{A}(\sigma_{i-2^{l+1}+1:i+1})$, which we denote as $o_i^{(l)} \sim \mathcal{A}(\sigma_{i-2^{l+1}+1:i+1})$. We will verify this assumption in §7.3.

If $o_i^{(l)} \sim \mathcal{A}(\sigma_{i-2^{l+1}+1:i+1})$ is true, Lemma 1 implies that RegularGPT's FFN models the transition function composition. This is immediate by setting $o_{i-2^l}^{(l-1)} \sim \text{Flat}(\mathcal{A}(\sigma_{i-2^{l+1}+1:i-2^l+1}))$, $o_i^{(l-1)} \sim \text{Flat}(\mathcal{A}(\sigma_{i-2^l+1:i+1}))$ and recognizing the fact that function composition is a matrix product under the representation of Eq. (2).

The next step is to explain the use of self-attention layers in RegularGPT. Although Lemma 1 has established a composition, it is unclear how the transitions are concatenated in the first place (i.e., $[\text{Flat}(A), \text{Flat}(B)]$). With a two-head self-attention and the learnable relative positional scalars, it is possible to adjust them so that the attention output contains the concatenated information $[\text{Flat}(A), \text{Flat}(B)]$.

Recall in Eq. (1), each head has a different set of scalars $r_i$'s. One concrete construction for concatenation is setting $r_0 = 0$ and the remaining $-\infty$ for the first head; $r_1 = 0$ and the remaining $-\infty$ for the second head. In other words, each head is only responsible for capturing one state transition. After the multi-head self-attention operation, we obtain the concatenation of two state transitions.

Finally, when the prediction head reads out the answer, the operation is equivalent to a mapping from $\mathcal{A}(\sigma_{0:T}) \in \mathbb{R}^{|Q| \times |Q|}$ to $\mathcal{A}_{q_0}(\sigma_{0:T}) = \mathcal{A}(\sigma_{0:T}) \circ q_0 \in \mathbb{R}^{|Q|}$. Since we assume that $o_{T-1}^{(l)}$

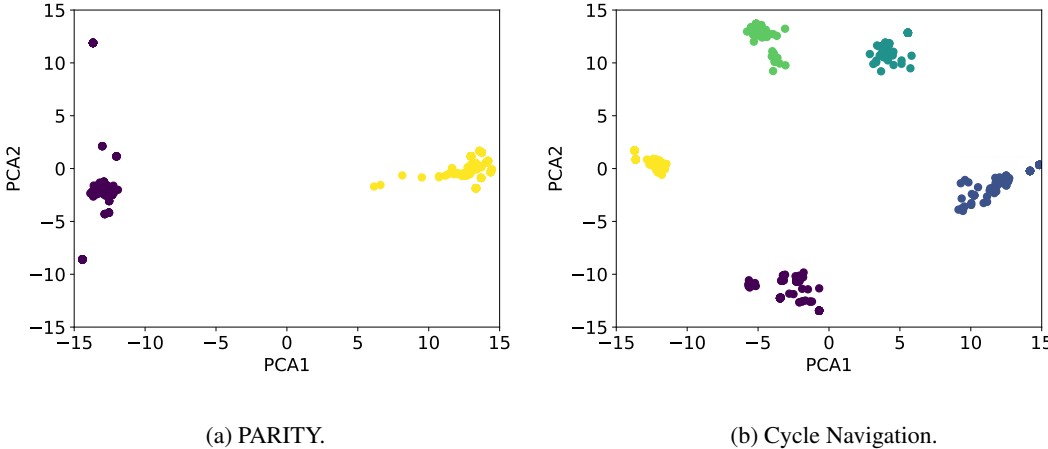

(a) PARITY.

(b) Cycle Navigation.

Figure 3: Clustering of FFN output vectors across all layers via PCA on the task of PARITY and Cycle Navigation.

models $\mathcal{A}(\sigma_{0:T})$, the transduction readout is performed by a linear map on $o_{T-1}^{(l)}$ as $W_o o_{T-1}^{(l)}$. Our tree-structured construction also guarantees that the final answer could be derived using $\log_2 T$ layers.

## 7.3 Verification of Transition Modeling

To verify whether our model learns the dynamics of a semiautomaton, we perform a clustering experiment to demystify the FFN output representations on the tasks of PARITY and Cycle Navigation. The two tasks are chosen as we can easily derive their state transition functions. For example, there are only two state transitions in PARITY:

$$\begin{bmatrix} 1 & 0 \\ 0 & 1 \end{bmatrix} \quad \text{or} \quad \begin{bmatrix} 0 & 1 \\ 1 & 0 \end{bmatrix}$$

and five state transitions in Cycle Navigation[5]:

$$\begin{bmatrix} \text{OneHot}_5((0+k)\bmod 5) \\ \text{OneHot}_5((1+k)\bmod 5) \\ \text{OneHot}_5((2+k)\bmod 5) \\ \text{OneHot}_5((3+k)\bmod 5) \\ \text{OneHot}_5((4+k)\bmod 5) \end{bmatrix}, \quad \text{for } k \in [0,...,4].$$

e.g., $k = 2$ gives $\begin{bmatrix} 0 & 0 & 1 & 0 & 0 \\ 0 & 0 & 0 & 1 & 0 \\ 0 & 0 & 0 & 0 & 1 \\ 1 & 0 & 0 & 0 & 0 \\ 0 & 1 & 0 & 0 & 0 \end{bmatrix}$.

Given a testing input sequence of length 500 that is much longer than the training length 40, we

extract the output $o_i^{(l)}$ of all layers $l$, perform dimension reduction using PCA, and plot the dimension-reduced points on a 2D plane. Ideally, we want to see a limited number of clusters across all layers, indicating the model learns to capture the state transition function. As we can see in Figure 3, PARITY has 2 clusters and Cycle Navigation has 5 clusters. The clear clustering effect demonstrates RegularGPT's correct learning of state transition functions. This is in contrast to the naive-summation approach learned by a vanilla Transformer as shown in Figure B.4 of Deletang et al. (2023).

## 7.4 Receptive Field Analysis

We resort to the gradient analysis tool (Chi et al., 2023) to inspect the receptive field of RegularGPT on regular and natural languages. It computes a cumulative sum of the gradient norms starting from the most recent token to the earliest one. A large magnitude of slope at a position means the most recent token has a high dependency on that position. Ideally, we would like to see the receptive field covering the whole input sequence for the case of regular languages because every single bit in the input sequence is important for the final results. This is equivalent to a slanted line going from the lower right to the upper left, which is validated in Figure 4a. As for natural language, we discover something interesting in Figure 4b in that RegularGPT settles on the local windowed-attention pattern as those enforced manually in prior work (Press et al., 2022; Chi et al., 2022, 2023). This suggests the task of natural language modeling mostly needs only local context to achieve good performance,

---

[5]Cycle Navigation (§5.1) has 3 one-step transitions (i.e., $|\mathcal{A}(\sigma_{a:a+1})| = 3$). Composing these transitions yields 5 different multi-step state transitions (i.e., $|\mathcal{A}(\sigma_{a:b})| = 5$ if $b - a \geq 5$).

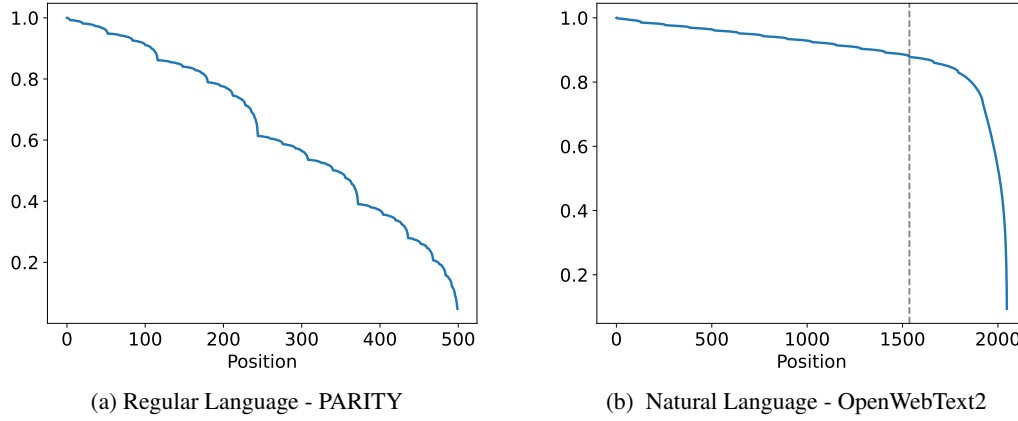

(a) Regular Language - PARITY  (b) Natural Language - OpenWebText2

Figure 4: Receptive field of RegularGPT via the cumulative gradient analysis tool (Chi et al., 2023).

which aligns with the common belief.

## 8 Conclusion

This paper introduces RegularGPT, a novel variant of the Transformer architecture inspired by the notion of working memory that can effectively model regular languages with high efficiency. Theoretical explanations and accompanying clustering visualizations are presented to illustrate how RegularGPT captures the essence of regular languages. Moreover, RegularGPT is evaluated on the task of natural language length extrapolation, revealing its intriguing rediscovery of the local windowed attention effect previously observed in related research. Notably, RegularGPT establishes profound connections with various existing architectures, thereby laying the groundwork for the development of future Transformer models that facilitate efficient algorithmic reasoning and length extrapolation.

## Limitations

Currently we set the chunk size $C$ of RegularGPT to a constant. Can we make the chunk size more flexible? A flexible and data-driven $C$ might further boost its performance on natural languages as they often demonstrate diverse patterns unlike regular languages underpinned by simple grammars. This might also improve the performance of RegularGPT when $C \neq 128$.

## Acknowledgment

We thank the anonymous reviewers for their insightful feedback and suggestions. We thank Princeton Research Computing for the technical support on the Della and Adroit clusters. The first author acknowledges the support from the Boeing Company (2019-STU-PA-259). The fourth author acknowledges the support from NSF. (MRI Award: 1919452).

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

## A  Hyperparameters for the Regular Language Experiments

We report the hyperpamaters used in the regular language experiments (Table 1) in Table 4.

## B  Hyperparameters for the Natural Language Experiments

We report the hyperpamaters used in the natural language experiments (Table 3) in Table 5.

## C  Proof of Lemma 1

**Lemma 1** (Approximation for Binary Matrix Product). *Let $A, B \in \{0,1\}^{n \times n}$ be binary matrices of dimension $n \times n$. Then, there exists a two-layer ReLU network such that*

$$f_{mlp}([Flat(A), Flat(B)]) = Flat(AB),$$

*where $Flat(X)_{(i-1)n+j} = X_{i,j}$ for $i, j \in [1, ..., n]$ is the operation that flattens a matrix into a vector.*

*Proof.* Observe that a ReLU operation can perfectly approximate the multiplication of two binary scalars:

$$\text{ReLU}(a + b - 1) = a \cdot b, \quad \text{for } a, b \in \{0, 1\}.$$

The binary matrix product $AB$ is composed of $n^3$ binary scalar products of the form:

$$A_{ik}B_{kj} = x_{(i-1)n+k}x_{(n+k-1)n+j}$$
$$\text{for } i, j, k \in [1, .., n],$$

where $x = [\text{Flat}(A), \text{Flat}(B)]$ is the concatenated flattened input. Our goal is to construct two neural network layers. The first layer computes all $n^3$ binary scalar products. The second layer sums these products into the form of matrix product; i.e., $\sum_{k=1}^{n} A_{ik}B_{kj}$.

The first layer's binary weight matrix $W^{(1)} \in \{0,1\}^{2n^2 \times n^3}$ is constructed as:

For $z \in [1, ..., 2n^2], \ i, j, k \in [1, ..., n]$,

$$W^{(1)}_{z,(i-1)n^2+(j-1)n+k} =$$

$$\begin{cases} 1 & \text{if } z = (i-1)n+k \text{ or } (n+k-1)n+j \\ 0 & \text{otherwise.} \end{cases}$$

$$(3)$$

Then, the first layer computes all $n^3$ binary scalar products as follows:

$$\text{ReLU}\left([\text{Flat}(A),\text{Flat}(B)]W^{(1)} - \mathbb{1}_{n^3}^{\top}\right)_{(i-1)n^2+(j-1)n+k}$$
$$= A_{ik}B_{kj} \quad \text{for } i, j, k \in [1, ..., n].$$

To sum these $n^3$ products into $n^2$ results, the second layer's binary weight matrix $W^{(2)} \in \{0,1\}^{n^3 \times n^2}$ is constructed as:

$$W^{(2)} = I_{n^2} \otimes \mathbb{1}_n = \begin{bmatrix} \mathbb{1}_n & 0_n & 0_n & \dots & 0_n \\ 0_n & \mathbb{1}_n & 0_n & \dots & 0_n \\ \vdots & & & & \vdots \\ 0_n & \dots & & 0_n & \mathbb{1}_n \end{bmatrix}$$

$$\in \{0,1\}^{n^3 \times n^2},$$

where $I_{n^2}$ is an $n^2 \times n^2$ identity matrix, $\otimes$ is the Kronecker product, $0_n$ is an n-dimensional column vector of all zeros, and $\mathbb{1}_n$ is an n-dimensional column vector of all ones. We arrive at a two-layer ReLU network that perfectly approximates the multiplication of two binary matrices:

$$f_{\text{mlp}}([\text{Flat}(A), \text{Flat}(B)])$$
$$= \text{ReLU}\left([\text{Flat}(A),\text{Flat}(B)]W^{(1)} - \mathbb{1}_{n^3}^{\top}\right)W^{(2)}$$
$$= \text{Flat}(AB).$$

$$\square$$

## D  Illustration of Lemma 1

### D.1  Illustration of the Binary Weight Matrices

We illustrate $W^{(1)}$ and $W^{(2)}$ of Lemma 1 as follows:

```python
import numpy as np

def get_W1(n):
    n2 = n*n
    W1 = np.zeros((2*n*n, n**3), dtype=int)
    for i in range(n):
        for j in range(n):
            for k in range(n):
                W1[i*n+k, i*n2+j*n+k] = 1
                W1[n2+k*n+j, i*n2+j*n+k] = 1
    return W1

def get_W2(n):
    eye = np.eye(n*n, dtype=int)
    ones = np.ones((n,1), dtype=int)
    W2 = np.kron(eye, ones)
    return W2
```

get_W1(2) gives:

```
[[1  0  1  0  0  0  0  0]
 [0  1  0  1  0  0  0  0]
 [0  0  0  0  1  0  1  0]
 [0  0  0  0  0  1  0  1]
 [1  0  0  0  1  0  0  0]
 [0  0  1  0  0  0  1  0]
 [0  1  0  0  0  1  0  0]
 [0  0  0  1  0  0  0  1]]
```

| # Layers | $\log_C T$ |
|---|---|
| Hidden Size | 256 |
| # Attention Heads | 8 |
| Train Seq. Len. | 40 or 50 |
| # Trainable Params. | 4.3 M |
| Optimizer | Adam (lr 1e-4, 3e-4, 5e-4) |
| Batch Size | 128 |
| Train Steps | 100,000 |
| Precision | float32 |
| Dataset | Regular Languages |

Table 4: Hyperparameters for the regular language experiments.

| # Layers | $K \log_C T$ |
|---|---|
| Hidden Size | 768 |
| # Attention Heads | 12 |
| Train Seq. Len. | 512 |
| # Trainable Params. | 81M ($K = 6$) or 123M ($K = 12$) |
| Optimizer | Adam (lr 6e-4) |
| Batch Size | 32 |
| Train Steps | 50,000 |
| Precision | bfloat16 |
| Dataset | OpenWebText2 |

Table 5: Hyperparameters for the natural language experiments.

get_W2(2) gives:

```
[[1  0  0  0]
 [1  0  0  0]
 [0  1  0  0]
 [0  1  0  0]
 [0  0  1  0]
 [0  0  1  0]
 [0  0  0  1]
 [0  0  0  1]]
```

### D.2 An Illustrative Example for $n = 2$

Suppose the input matrices are:

$$A = \begin{bmatrix} 1 & 0 \\ 1 & 0 \end{bmatrix}, \quad B = \begin{bmatrix} 0 & 1 \\ 1 & 0 \end{bmatrix}.$$

The concatenated flattened input becomes:

$$x = [\text{Flat}(A), \text{Flat}(B)] = [1\ 0\ 1\ 0\ 0\ 1\ 1\ 0].$$

Then, Lemma 1 is verified as follows:

$$\text{ReLU}\left(xW^{(1)} - \mathbb{1}_{n^3}^\top\right) W^{(2)}$$
$$= \text{ReLU}\left([1\ 1\ 2\ 0\ 1\ 1\ 2\ 0] - 1\right) W^{(2)}$$
$$= [0\ 0\ 1\ 0\ 0\ 0\ 1\ 0] W^{(2)}$$
$$= [0\ 1\ 0\ 1]$$
$$= \text{Flat}\left(\begin{bmatrix} 0 & 1 \\ 0 & 1 \end{bmatrix}\right) = \text{Flat}\left(AB\right).$$

Here is the Python code for the above example:

```python
A = np.array([[1,0],[1,0]]).reshape(-1)
B = np.array([[0,1],[1,0]]).reshape(-1)
x = np.concatenate([A, B]).reshape(1,-1)
W1 = get_W1(2)
W2 = get_W2(2)
flat_AB = np.maximum(x @ W1 -1,0) @ W2
```