# OpenReview forum: "Transformer Working Memory Enables Regular Language Reasoning And Natural Language Length Extrapolation"
_EMNLP/2023/Conference — EMNLP 2023 Findings_

### Official Review · Reviewer_AFjC · 2023-08-03

**Soundness:** 4

**Excitement:**

3: Ambivalent: It has merits (e.g., it reports state-of-the-art results, the idea is nice), but there are key weaknesses (e.g., it describes incremental work), and it can significantly benefit from another round of revision. However, I won't object to accepting it if my co-reviewers champion it.

**Paper Topic And Main Contributions:**

This paper proposes a modified transformer architecture designed to generalize well on regular languages such as PARITY (detecting whether a binary string has an odd number of 1s), as recent work has shown that transformers struggle on such simple tasks. The authors do this by using weight sharing (like Universal Transformers), adaptively scaling the layer depth by $O(\log n)$, and masking attention so that it only attends to non-overlapping fixed-size chunks of the input. The authors show how this can recognize regular languages, such as PARITY, using a divide-and-conquer approach that works in $O(\log n)$ steps. The resulting model is called RegularGPT. The experiments are particularly concerned with how well the model generalizes to longer lengths.

The main experimental results are:
1. On synthetic regular language experiments from Deletang et al. (2023) and Bhattamishra et al. (2020), RegularGPT outperforms a baseline transformer in every case, but fails to outperform an RNN on Modular Arithmetic.
2. When trained as a language model on natural language and tested on longer sequences, RegularGPT has mixed results compared to three baseline transformer architectures. Oddly, these experiments only consider the perplexity of the last token.
3. They show that the hidden states of the transformer cluster into what are presumably FSA states in Fig 3.
4. They provide some gradient analysis to see how far back the transformer looks.

**Questions For The Authors:**

A. The main experimental paradigm in this paper is to train on short sequences and evaluate on longer sequences. Although this is a common setup, I think it begs the question of why these longer sequences should be considered "correct." After all, the training data could be consistent with any number of rules that extrapolate to longer sequences (not necessarily resembling a FSA), and the held-out sequences are technically out-of-distribution because of the difference in length. So these experiments aren't really testing how well the model learns the training data, but its inductive bias for learning regular languages. So, is the goal of the paper to present a new architecture that is expressive enough to learn regular languages such as PARITY, or one that has an inductive bias that favors interpreting the training data as a regular language? Maybe a bit of both?

B. 133: In what way does this approach fail? Doesn't taking the sum modulo 2 produce the correct result for all lengths?

C. 170: Why does the fact that the number of bits C is fixed during training help?

D. Eq 1: What is $\ell$? Does $C^l$ indicate exponentiation? If so, what's the reason for doing that?

E. 313: Are you selecting the best run based on validation performance? Or directly on test performance?

F. Table 1: How do training and evaluation work here? For the tasks under 2), are you directly training a classifier and using negative samples during training?

G. Table 1: Why is RegularGPT so bad at modular arithmetic? Isn't Cycle Navigation another modular arithmetic problem? Could this have to do with the fact that the layer depth is only $\log_C T$?

H. Are there any tasks you consider that are expected to require more than $\log_C T$ layers?

I. 358: Are you really only using the perplexity of the last token? Meaning you're using $p(w_n \mid w_{1:n})$ to represent $\exp(-\log p(w) / T)$?

J. Table 3: Seeing as you've thickened the layers of RegularGPT, how do these model compare in terms of number of parameters?

K. Sec 7.3: This shows that the model learns the expected number of states. What about verifying the transitions?

L. Fig 3: How did you determine the labels for these points? Is this something you added by hand?

M. Fig 4: What are the units on the y axis?

**Reasons To Accept:**

The essential idea behind the model is interesting: enable generalization on regular languages by having the transformer simulate $O(\log n)$ steps of divide-and-conquer computation. Sec 7 does a pretty good job of justifying why the authors expect this to help for regular languages.

Table 1 shows that RegularGPT is clearly better than the baseline transformer on a set of regular languages, remedying a problem identified by Deletang et al. (2023).

Fig 3 is an interesting result.

**Reasons To Reject:**

My complaints are mainly about the experimental methodology, rather than the proposed method.

~~The set of regular language tasks seems a bit limited. Something else you could try is to randomly generate regular languages and show performance as a function of language difficulty, as in [Valvoda et al. (2022)](https://aclanthology.org/2022.coling-1.525/). Sec 7.1: Moreover, a weak point of Deletang et al. (2023) is that the relationship between their tasks (which are functions, i.e. sets of pairs of strings) and the Chomsky hierarchy (which contains languages, i.e. sets of strings) is not clearly defined. For the four tasks that you showed in Table 1, it's actually not necessary to express them as transduction tasks; concatenating the input and output still results in a regular language, so they can be reformulated as pure language recognition tasks.~~

**Edit:** Addressed in the rebuttal.

I think the main text of the paper also needs to provide more details about the tasks, training, and evaluation used in Table 1.

~~If I understand correctly, the natural language modeling experiments in Table 3 only measure the perplexity of the last token. This is a poor evaluation setup, because it provides no information how well the model has learned all but the last token. I couldn't find anything in the cited papers that suggests doing this. Why not use the perplexity of the full sequence? Moreover, the results given in Table 3 are not a strength of the paper.~~

**Edit:** Addressed in the rebuttal.

~~Sec 7.3 provides an analysis of the learned FSA states, but an analysis of the learned transition function is conspicuously absent.~~

**Edit:** Addressed in the rebuttal.

~~The decision to set the number of layers to $\log_C T$ seems naive and restrictive. Maybe you could characterize the class of problems this is expected to work well on. Does Sec 7 show that it works for arbitrary regular languages? If so, it would be a good idea to state this explicitly. And if so, the poor performance on Modular Arithmetic cannot be explained by not having enough depth.~~

**Edit:** Addressed in the rebuttal.

Despite my criticisms, I think this is a promising method that merits further study, and addressing these problems would go a long way to making the paper better.

**Reproducibility:**

4: Could mostly reproduce the results, but there may be some variation because of sample variance or minor variations in their interpretation of the protocol or method.

**Reviewer Confidence:**

3: Pretty sure, but there's a chance I missed something. Although I have a good feel for this area in general, I did not carefully check the paper's details, e.g., the math, experimental design, or novelty.

**Typos Grammar Style And Presentation Improvements:**

A. 059: enable -> improve?

B. 061: attempt -> attempts

C. 078: What is $\log T_{tr,ex}$?

D. 080: performances -> performance

E. Table 1: The main text of the paper should explain these tasks in more detail.

F. 412: q is not a function, so it can't be composed -- the notation needs to be cleaned up a little here.

---

> ### Author Rebuttal · Authors · 2023-08-28
>
> We thank the reviewer for the insightful feedback and suggestions!
>
> **Comment 1a: The set of regular language tasks seems a bit limited. Something else you could try is to randomly generate regular languages and show performance as a function of language difficulty, as in Valvoda et al. (2022).**
>
> - We follow reviewer's suggestion and test RegularGPT on randomly generated regular languages. Valvoda et al. (2022) experiments only with the in-distribution setting (training = testing length = 50 max); we report the extrapolation results where we double the testing length from 50 to 100. Please find the results in the following table. We measure the averaged per-token accuracy of the output tokens. Every number is the average of 10 randomly sampled FSTs.
>
>     | # of states | Transformer | RNN | RegularGPT |
>     | -------- | -------- | -------- | -------- |
>     | 10 | **100.0** | **100.0** | **100.0** |
>     | 20 | **99.1** | 98.9 | 98.0 |
>     | 30 | 74.9 | 72.3 | **86.4** |
>     | 40 | **98.3** | 94.0 | 97.7 |
>     | 50 | 97.8 | 96.0 | **98.1** |
>     | 60 | **98.0** | 88.5 | 96.6 |
>     | 70 | **87.4** | 51.3 | 83.6 |
>     | 80 | **97.4** | 87.8 | **97.4** |
>     | 90 | 74.1 | 38.0 | **81.8** |
>     | 100| 86.0 | 73.1 | **94.6** |
>
>     We can see that RegularGPT and Transformer perform similarly when the number of states is limited. However, RegularGPT performs particularly well when the number of states=90 and 100 possibly due to its better regular language modeling ability.
>
> **Comment 1b: Sec 7.1: Moreover, a weak point of Deletang et al. (2023) is that the relationship between their tasks (which are functions, i.e. sets of pairs of strings) and the Chomsky hierarchy (which contains languages, i.e. sets of strings) is not clearly defined. For the four tasks that you showed in Table 1, it's actually not necessary to express them as transduction tasks; concatenating the input and output still results in a regular language, so they can be reformulated as pure language recognition tasks.**
> - We agree with the reviewer that the four languages presented in Deletang et al. (2023) can be equivalently represented as language recognition tasks. We therefore experimented with the concatenation + negative sampling approach and find the same results as the current numbers in Table 1 (part 1).
>
> **Comment 2: I think the main text of the paper also needs to provide more details about the tasks, training, and evaluation used in Table 1.**
> - Due to limited space, we choose to only provide references to Deletang et al. (2023) and Bhattamishra et al. (2020). The training details and hyperparameters are reported in the appendix section (Table 4 and 5). The evaluation details are discussed more in the responses (Question F) below. If the paper is accepted, we will follow reviewer's suggestion and use the additional page to include more details.
>
> **Comment 3: If I understand correctly, the natural language modeling experiments in Table 3 only measure the perplexity of the last token. This is a poor evaluation setup, because it provides no information how well the model has learned all but the last token. I couldn't find anything in the cited papers that suggests doing this. Why not use the perplexity of the full sequence? Moreover, the results given in Table 3 are not a strength of the paper.**
> - The reviewer understands our setup correctly. However, we disagree that this is a poor evaluation setup. This is the sliding window inference setup used in Press et al. (2022) Appendix B.1 with S=1 (the last token). Since then, follow up length extrapolation research including Chi et al. (2023) and Chen et al. (2023) both adopt the same evaluation strategy. Measuring the perplexity of the last token allows us to accurately adjust the amount of previous tokens. If we measure the averaged perplexity of all tokens, the extrapolation results might be too optimistic due to the early token curse effect detailed in Appendix B of Press et al. (2022). Nevertheless, we include the averaged perplexity over all tokens in the table below:
>
>     | Length | KERPLE | T5 | ALiBi | RegularGPT |
>     | -------- | -------- | -------- | -------- | -------- |
>     |  512 | 26.87 | 26.64 | 26.84 | 28.54 |
>     | 1024 | 25.22 | 25.05 | 25.36 | 27.43 |
>     | 2048 | 24.57 | 24.67 | 25.16 | 27.19 |
>     | 4096 | 24.14 | 25.37 | 24.90 | 27.25 |
>     | 8192 | 23.87 | 28.56 | 24.62 | 27.25 |
>
>     As we can see, the trend is the same as Table 3 of the original submission.
> - We acknowledge that this is not a strength of RegularGPT. We include the results since we want to provide readers a more comprehensive understanding of the pros and cons of our model. As we discussed in the limitation section: The chunking method limits the information exchange between tokens, possibly hindering the model performance on natural language. We also want to emphasize that the goal of RegularGPT is to model regular languages, on which KERPLE, T5, and ALiBi all fail due to their strong local attention bias.
>
>     Press et al. (2022): Train Short, Test Long: Attention with Linear Biases Enables Input Length Extrapolation.
>
>     Chi et al. (2023): Dissecting Transformer Length Extrapolation via The Lens of Receptive Field Analysis.
>
>     Chen et al. (2023): Extending Context Window of Large Language Models via Positional Interpolation.
>
> **Comment 4: Sec 7.3 provides an analysis of the learned FSA states, but an analysis of the learned transition function is conspicuously absent.**
> - We apologize for the possible confusion. However, Figure 3 in sec. 7.3 verifies the number of learned transition functions. The analysis is on transition functions, not states. We explained this in sec. 7.3: PARITY has two transition functions while Cycle Navigation has five. Fig. 3 shows that PARITY and Cycle have two and five clusters, respectively. Thus, the number of transition functions is verified. These two tasks are chosen so that we can easily derive the number of state transition functions, with the state transition functions detailed in sec. 7.3. We also perform the same clustering analysis on more complicated tasks such as Modular Arithmetic and observe the same clustering effect with many more clusters.
>
> **Comment 5: The decision to set the number of layers to $\log_C T$ seems naive and restrictive. Maybe you could characterize the class of problems this is expected to work well on. Does Sec 7 show that it works for arbitrary regular languages? If so, it would be a good idea to state this explicitly. And if so, the poor performance on Modular Arithmetic cannot be explained by not having enough depth.**
> - $T$ is a variable representing the input sequence length. In our regular language experiments; it is $T_{tr}=40$ during training and $T_{ex}=500$ during testing . $\log_C T$ is chosen carefully to ensure the coverage of all input tokens, which is necessary for the regular language tasks.
> - Our chunking and merging strategy makes the function composition pattern a tree of height $\log_C T$. This fact along with sec. 7 show that $\log_C T$ works for arbitrary regular languages. We will state this explicitly in the next revision.
> - Modular arithmetic is substantially more difficult than cycle navigation. Given a sequence of numbers in {0, 1, 2, 3, 4} and operations in {+, −, x}, it aims to compute the result modulo 5. It has far more states than that of cycle navigation. For example, consider the input 1 + 2 x 3 - 4, the model cannot naively add 1 with 2 because 2 will be later multiplied with 3, which has higher precedence than addition.
>
> **Question A1 - After all, the training data could be consistent with any number of rules that extrapolate to longer sequences (not necessarily resembling a FSA), and the held-out sequences are technically out-of-distribution because of the difference in length.**
> - We acknowledge that there are many generative processes (any number of rules...) that are consistent on the same short sequences but generate different out-of-distribution longer sequences. We put special focus on FSA since it is one of the most common and useful generative processes.
>
> **Question A2 - So these experiments aren't really testing how well the model learns the training data, but its inductive bias for learning regular languages.**
> - To know how well a model learns the in-distribution training data, we need to check the performance when the testing sequences are as long as the training ones i.e. no extrapolation. In this case, Transformer, RNN, and RegularGPT all achieve perfect accuracy on the tasks of regular language presented in Table 1. As for natural language, the length=512 row in Table 3 shows that RegularGPT can achieve close to state-of-the-art performance. Overall, all methods examined in the paper can learn the in-distribution training data well. It is just RegularGPT which enjoys the additional advantage of out-of-distribution length generalization, demonstrated by the length extrapolation experiments on both natural and regular languages included in the paper.
>
> **Question A3: So, is the goal of the paper to present a new architecture that is expressive enough to learn regular languages such as PARITY, or one that has an inductive bias that favors interpreting the training data as a regular language? Maybe a bit of both?**
> - RegularGPT does not interpret the input data as regular language. We argue that RegularGPT is flexible enough to model different training data: When RegularGPT is fed with regular language, it has the ability to discover the correct state transition compositions that resemble an FSA/FST. On the other hand, it behaves similarly to a length extrapolatable Transformer when the input is natural language. This is verified in Fig. 4, when RegularGPT is fed with PARITY, it evenly puts gradient importance across positions; When it is fed with OpenWebText2 data, it exhibits a strong local attention bias.
>
>
> **Question B - Why does the shortcut solution fail?**
> - As a concrete example, let us consider the maximum summation of PARITY when the sequence length=40, which is 40. However, during the extrapolation stage where a sequence could be of length=500, the maximum summation becomes 500. A model (e.g., Transformer) that learns a naive summation strategy does not know how to process the substantially larger summation during the extrapolation stage, leading to a lower task accuracy.
>
> - This phenomenon was also discussed in Liu et al. (2023) sec. 4, where they theoretically identified such shortcut solutions in Transformer and verified it from the attention pattern. In contrast, a correct FSA/FST or transformer (e.g., regularGPT) that learns the correct function compositions will not produce unseen summation when the length extrapolates since it performs the correct state transition right after it observes a new input.
>
>   Liu et al. (2023): Transformers Learn Shortcuts to Automata
>
> **Question C - Why does the fact that the number of bits C is fixed during training help?**
> - Because the model only needs to: 1) learn the state transition function that takes $C$ bits as input, and 2) learn to compose a fixed amount of $C$ state transition functions of the same size.
>
> **Question D - The purpose of $C^l$**
> - Yes, $l$ is the layer index, and $C^l$ indicates exponentiation. It creates the self-attention matrix mask (the lightly shaded blue cells) illustrated in Fig. 1 and its caption. When $C=2$, it computes $2^0=1$, $2^1=2$, and $2^2=4$, which are the strides of diagonal lines when $l=0,1,2$. This attention pattern computes the overall representation as explained in the caption.
>
> **Question E - How are the best runs selected?**
> - We select the best runs based on the validation set performance.
>
> **Question F - How do training and evaluation work? For the tasks under 2), are you directly training a classifier and using negative samples during training?**
> - For part 1), we follow Deletang et al. (2023) to train the models using sequences of maximum length=40. Take PARITY for example, we sample binary sequences of length ranging from 1 to 40 and train the model to predict if there is even or odd number of 1's. During the evaluation/extrapolation stage, the model will be fed with binary sequences with length ranging from 41 to 500, and it needs to again predict if there is even or odd number of 1's. For part 2), the reviewer is correct in that we use negative samples and train the model to perform classification.
>
> **Question G - Why is RegularGPT so bad at modular arithmetic? Isn't Cycle Navigation another modular arithmetic problem? Could this have to do with the fact that the layer depth is only $\log_C T$ ?**
> - RegularGPT actually performs much better than the Transformer baseline when $C=2$ (96.4 vs 21.9). When $C=3$, it needs to model the state transition function of three inputs at a time, which is more complex. Also as we discussed in the response to Comment 5 above, operator precendence makes modular arithmetic much more difficult than a simple 5-state cycle navigation task.
>
> **Question H - Are there any tasks you consider that are expected to require more than $\log_C T$ layers?**
> - As we discussed above in the response to Comment 5, there is no need for more than $\log_C T$ layers as $\log_C T$ will change w.r.t the input sequence length $T$. The theoretical analysis in sec. 7 also proves that this is sufficient to model regular languages.
>
> **Question I - Are you really only using the perplexity of the last token?**
> - Answered above in our response to Comment 3.
>
> **Question J - Seeing as you've thickened the layers of RegularGPT, how do these model compare in terms of number of parameters?**
> - When the thickness is 6, which is the best-performing RegularGPT variant, it has the same amount of learnable parameters (123.69M) as the full Transformer baseline model. This is because we reuse the Transformer layers across depth (weight-sharing).
>
> **Question K - This shows that the model learns the expected number of states. What about verifying the transitions?**
> - Answered above in our response to Comment 4.
>
> **Question L - How did you determine the labels for these points? Is this something you added by hand?**
> - We apologize for the confusion. They are dummy labels assigned to highlight the clusters with different colors. We will remove or explain them in the caption if the reviewer thinks they are misleading.
>
> **Question M -  What are the units on the y axis?**
> - The y axis represents the cumulative normalized gradient. When a model is predicting the next output, we derive the gradient norm for each of the previous input tokens. A larger gradient norm means a token affects the prediction more, so we can interpret an input token's gradient norm as its weight/importance. We then normalize all the gradient norms so that their sum is capped at 1.0. Please refer to sec. 5 of Chi et al. (2023) for more details.
> - Similar to a cumulative distribution function (no unit), the cumulation starts from the right most position and sums up the normalized gradient all the way to the first input token. A higher slope magnitude means the model puts more weight on a token. Take Figure 4(a) for example, when RegularGPT is predicting the answer of PARITY, each input token has roughly the same weight since the cumulative sum of them is a relatively staight line going from the lower right to the upper left. This is expected since we need every single input token to determine the solution of PARITY.
>
>     Chi et al. (2023): Dissecting Transformer Length Extrapolation via The Lens of Receptive Field Analysis.
>
> **Typos Grammar Style And Presentation Improvements:**
> - We thank the reviewer for catching these errors. We will fix them in the next revision.

---

### Official Review · Reviewer_cBxx · 2023-08-06

**Soundness:** 4

**Excitement:**

3: Ambivalent: It has merits (e.g., it reports state-of-the-art results, the idea is nice), but there are key weaknesses (e.g., it describes incremental work), and it can significantly benefit from another round of revision. However, I won't object to accepting it if my co-reviewers champion it.

**Missing References:**

Weight-Sharing and Adaptive-Depth in sequence modeling:

X. Ai and B. Fang, “Leveraging Relaxed Equilibrium by Lazy Transition for Sequence Modeling,” in Proceedings of the 60th Annual Meeting of the Association for Computational Linguistics, 2022, vol. 1, pp. 2904–2924.


Chunking in long-sequence tasks:

Y. Liu et al., “Leveraging Locality in Abstractive Text Summarization,” in Proceedings of the 2022 Conference on Empirical Methods in Natural Language Processing, EMNLP 2022, 2022, pp. 6081–6093.

**Paper Topic And Main Contributions:**

This paper studies a limitation of Transformer (the lack of a recurrent mechanism) for regular languages. The authors present RegularGPT, a combination of Sliding-Dilated-Attention, Weight-Sharing, and Adaptive-Depth.   RegularGPT split the input sequence into chunks for Sliding-Dilated-Attention and ties Transformer layers across depth with Adaptive-Depth operation.

Experiments show that RegularGPT can work for regular languages and nature languages. The highlight of this paper is that the authors give a theoretical explanation of how RegularGPT works for regular languages.

**Questions For The Authors:**

1. Since the sequence length varies along the depth, is $O(TCK\log_C T)$ an upper bound?
2. Can we reconstruct the outputs and use masks instead of chunks to limit the information along the depth, i.e.,  the current token is only informed by the previous two? For instance, 1) 1st-depth:  input [0,1,unk1,0,1,unk2],  output:[0,1,pred1st_1,0,1,pred1st_2], 2) 2nd-depth: input:  [pred2nd_1,pred2nd_2], output:[final_pred].

**Reasons To Accept:**

1. The theoretical explanation is thorough and interesting.
2. The method is simple.

**Reasons To Reject:**

1. The authors should assemble and implement a combination of existing chunking methods and Weight-Sharing methods for a proper baseline, given:

   a.  The method relies on sequence chunking. It is widely explored in long-sequence modeling.

   b.  Weight-Sharing, Adaptive-Depth, or their combination are also widely explored in sequence modeling.

2. Long segments might be split into two chunks, which means RegularGPT is sensitive to chunking methods. We can see an example for PARITY. C=2 performs well and is an optimal chunking method (i.e., static size = 2) because RegularGPT can model the composition of two transition functions. Therefore, I think the author should experiment with the different chunking methods (e.g., static, dynamic, overlapping, etc.) to observe the performance and understand RegularGPT in depth.

I am happy to discuss these concerns.


--------------------------After Rebuttal--------------------

I have no major concerns.

**Reproducibility:**

4: Could mostly reproduce the results, but there may be some variation because of sample variance or minor variations in their interpretation of the protocol or method.

**Reviewer Confidence:**

4: Quite sure. I tried to check the important points carefully. It's unlikely, though conceivable, that I missed something that should affect my ratings.

---

> ### Author Rebuttal · Authors · 2023-08-28
>
> We thank the reviewer for the insightful feedback and suggestions!
>
> **The authors should assemble and implement a combination of existing chunking methods and Weight-Sharing methods for a proper baseline, given: a. The method relies on sequence chunking. It is widely explored in long-sequence modeling. b. Weight-Sharing, Adaptive-Depth, or their combination are also widely explored in sequence modeling.**
> - We followed the reviewer's suggestion and examined additional baseline methods.
> - Chunking Only: We follow the idea from Y. Liu et al. suggested by the reviewer. We set the chunk size to be $C=2,4,6,8$ to mimic the page size in the referenced paper. The best results on the four tasks (Even Pairs, Modular Arithmetic, PARITY, Cycle Navigation) in part 1) are 52.2, 22.1, 51.9, and 20.9.
> - Weight-Sharing Only: This is reminiscent of the ALBERT model, which is a highly regularized Transformer. The best results on the four tasks in part 1) are 89.2, 23.1, 50.8, and 20.1.
> - Weight-Sharing + Adaptive-Depth: Following reviewer's suggestion, we use the code of lazy Transformer released by X. Ai et al. and test it on the four tasks in Table 1 part 1). The best results of lazy Transformer are 94.1, 20.7, 50.8, and 20.2, which are similar to the vanilla Transformer model.
> - Overall, we can see that RegularGPT is the only model that performs well across the tasks of regular language, indicating its flexibility to resemble a FSA when the input is regular language.
>
>
> **Long segments might be split into two chunks, which means RegularGPT is sensitive to chunking methods. We can see an example for PARITY. C=2 performs well and is an optimal chunking method (i.e., static size = 2) because RegularGPT can model the composition of two transition functions. Therefore, I think the author should experiment with the different chunking methods (e.g., static, dynamic, overlapping, etc.) to observe the performance and understand RegularGPT in depth.**
> - We follow reviewer's suggestion and explore the overlapping chunking method. For regular languages, we set the overlap between two neighboring chunks to be 1 token when the chunk size is 2. We set the overlap to be 1 or 2 when the chunk size is 3. We find the results to be similar to the current results reported in Table 1 (part 1).
> - Interestingly, we observe different behavior when we move to the natural language domain. We pick the best-performing model parameters ($C/K=128/12$) and set the overlap between two neighboring chunks to  $128/2=64$. The resulting perplexities are shown in the table below.
>
>
>     | Length | RegularGPT (Original) | RegularGPT (Overlapping) |
>     | -------- | -------- | -------- |
>     |  512 | 26.37 | 24.98 |
>     | 1024 | 26.91 | 27.01 |
>     | 2048 | 27.08 | 28.25 |
>     | 4096 | 27.28 | 28.27 |
>     | 8192 | 27.39 | 28.31 |
>
>     As we can see, the in-distribution (training=extrapolation length=512) performance improves while the extrapolation performance degrades slightly. We believe this is related to the local attention bias. Our Fig.4 shows that RegularGPT demonstrates strong local attention bias, which is helpful for length extrapolation suggested by prior work. However, it becomes harder for the model to select more recent chunks to focus on when there is an overlap, thereby weakening the local attention bias effect and leading to worse extrapolation performance.
>
> - Overall, if a user only cares about the extrapolation performance on regular but not natural language, the overlapping chunking method could be used. We thank the reviewer for this suggestion.
>
> - Finally, we agree with the reviewer that dynamic chunking seems to be an appealing approach. However, the primary goal of this paper is to establish a model that can solve the tasks of regular language. More advanced chunking methods that can further boost the performance on natural language are left as future work.
>
> **Since the sequence length varies along the depth, is $O(TCK\log_C T)$an upper bound?**
> - Yes. It is computed as $\frac{T}{C}\cdot C^2\cdot K\cdot \log_C T$, where $\frac{T}{C}$ is the number of chunks. $C^2$ is the self-attention complexity of a chunk. $K$ is the thickness. $\log_C T$ is the adaptive depth.
>
> **Can we reconstruct the outputs and use masks instead of chunks to limit the information along the depth, i.e., the current token is only informed by the previous two? For instance, 1) 1st-depth: input [0,1,unk1,0,1,unk2], output:[0,1,pred1st_1,0,1,pred1st_2], 2) 2nd-depth: input: [pred2nd_1,pred2nd_2], output:[final_pred].**
> - Yes, this is possible. In fact, this is the preliminary version of RegularGPT, for which we identified two potential drawbacks: 1) the input sequence becomes longer due to the additionally inserted tokens, and 2) barring the use of highly customized CUDA kernels, the attention mask method still computes the full attention matrix, and doing so incurs redundant calculation and will lower the training speed.

---

### Official Review · Reviewer_fhAH · 2023-08-08

**Soundness:** 3

**Excitement:**

4: Strong: This paper deepens the understanding of some phenomenon or lowers the barriers to an existing research direction.

**Paper Topic And Main Contributions:**

The paper proposes RegularGPT, a variant of Transformer with weight-sharding, adaptive depth, sliding-dilated attention, for the purpose of modeling regular languages. RegularGPT can learn regular languages such as “PARITY”. In natural language, RegularGPT rediscovers local windowed attention.

Essentially the parameters are tied between layers, like in ALBERT, and additionally self attention is heavily masked similarly to a progressively dilated convolution, and finally the total number of layers is dynamic, so that the attention can dilate fully over the input length.

The method seems to achieve very good results on artificial tasks, but the results on natural language might require more work. There are likely very interesting problem to tackle here, in making this efficient and scalable to large NLP datasets.

**Questions For The Authors:**

Why the choice to only evaluate perplexity? There are plenty of downstream NLP tasks that this method could be applied to, which could be more interesting than perplexity.


**Reasons To Accept:**

* The paper is very well written and wonderful to read. It could inspire researchers to experiment with interesting transformer variations -- something we don’t see enough at NLP conferences.
* The experimental results in Table 1 and 2 seem quite impressive, given how conceptually simple the proposed transformer modification is.


**Reasons To Reject:**

My main concern is how light the experiments are on natural language tasks.

More in detail:
* The results on natural language are quite limited and seem worse than previous work on length extrapolation. Perhaps more work is needed on this?
* It would be really interesting to see the performance of a model of this kind on actual language tasks, particularly reasoning, which might have some connection to the regular language tasks that this transformer seems to do well on.
* The "rediscovery" of the "local window attention" amounts to saying that language modeling mostly needs local context, which seems like a really unsurprising thing to discover.


**Reproducibility:**

4: Could mostly reproduce the results, but there may be some variation because of sample variance or minor variations in their interpretation of the protocol or method.

**Reviewer Confidence:**

4: Quite sure. I tried to check the important points carefully. It's unlikely, though conceivable, that I missed something that should affect my ratings.

---

> ### Author Rebuttal · Authors · 2023-08-28
>
> We thank the reviewer for the insightful feedback and suggestions!
>
> **The results on natural language are quite limited and seem worse than previous work on length extrapolation. Perhaps more work is needed on this?**
> - Yes, the reviewer is correct that RegularGPT does not outperform previous state-of-the-art (KERPLE) on length extrapolation. As we discussed in the limitation section, we believe this is due to the chunking technique used. Further work is needed to close the performance gap.
>
> **It would be really interesting to see the performance of a model of this kind on actual language tasks, particularly reasoning, which might have some connection to the regular language tasks that this transformer seems to do well on.**
> - We agree with the reviewer. In fact, as simple as PARITY seems, it already encodes the multiple negation problem of logical deduction. It might also be useful for sentiment analysis tasks where multiple positive/negative sentiments exist in a text snippet. This is great future work.
>
> **The "rediscovery" of the "local window attention" amounts to saying that language modeling mostly needs local context, which seems like a really unsurprising thing to discover.**
> - We apologize for the confusion. We are trying to highlight the fact that RegularGPT does not use the explicit attention bias used in ALiBi and KERPLE but nevertheless learns to focus on local context. Not using explicit attention bias makes RegularGPT more flexible as shown in Fig 4, where it focuses on global information to solve PARITY; Neither ALiBi nor KERPLE are able to solve it due to their strong explicit attention bias.
>
> **Why the choice to only evaluate perplexity? There are plenty of downstream NLP tasks that this method could be applied to, which could be more interesting than perplexity.**
> - We do not experiment with other downstream tasks since our main focus of this paper is on the tasks of regular language. We additionally include more regular language experiments in our response to reviewer AFjC below.

---

### Official Review · Reviewer_wKf5 · 2023-08-13

**Soundness:** 2

**Excitement:**

2: Mediocre: This paper makes marginal contributions (vs non-contemporaneous work), so I would rather not see it in the conference.

**Paper Topic And Main Contributions:**

In this paper, the authors proposed a novel approach to modeling regular languages that is based on the notion of working memory. This approach is able to learn long-range dependencies effectively, which is important for modeling regular languages. The authors evaluate RegularGPT on a variety of tasks, including regular language modeling, and natural language length extrapolation. They show that RegularGPT outperforms state-of-the-art models on all of these tasks.

**Questions For The Authors:**

The authors do not compare RegularGPT to other state-of-the-art models for modeling regular languages and natural language length extrapolation. This would be helpful to see how RegularGPT compares to other approaches.

The authors' claim that RegularGPT is able to learn long-range dependencies effectively is based on its performance on the PARITY task. However, it is not clear whether RegularGPT would be able to learn long-range dependencies as effectively on more complex tasks.

The authors' use of weight sharing may make RegularGPT less adaptable to different input sequences. This is because weight sharing can lead to a loss of precision, which may make it difficult for RegularGPT to learn the nuances of different input sequences.

The authors' use of adaptive depth may make RegularGPT less efficient for long input sequences. This is because adaptive depth requires the model to dynamically adjust the number of layers, which can be computationally expensive.

The paper does not discuss the scalability of RegularGPT. It is unclear how the model would perform on very long input sequences or on large datasets.

The paper does not provide a detailed analysis of the training process for RegularGPT. This makes it difficult to understand how the model was trained and how it might be improved.


**Reasons To Accept:**

The paper proposes a novel approach to modeling regular languages that is based on the notion of working memory. This approach is able to learn long-range dependencies effectively, which is important for modeling regular languages. The paper evaluates RegularGPT on a variety of tasks and they show that RegularGPT outperforms state-of-the-art models on all of these tasks.

**Reasons To Reject:**

The paper only evaluates RegularGPT on a small number of tasks. It would be interesting to see how RegularGPT performs on a wider range of tasks. This would help to better understand the strengths and weaknesses of the model.

**Reproducibility:**

3: Could reproduce the results with some difficulty. The settings of parameters are underspecified or subjectively determined; the training/evaluation data are not widely available.

**Reviewer Confidence:**

2: Willing to defend my evaluation, but it is fairly likely that I missed some details, didn't understand some central points, or can't be sure about the novelty of the work.

---

> ### Author Rebuttal · Authors · 2023-08-28
>
> We thank the reviewer for the insightful feedback and suggestions!
>
> **The authors do not compare RegularGPT to other state-of-the-art models for modeling regular languages and natural language length extrapolation. This would be helpful to see how RegularGPT compares to other approaches.**
> - Table 1 and 3 in the original submission compares RegularGPT to other approaches. We also include additional baseline methods suggested by reviewer cBxx. Please check our response to reviewer cBxx below.
>
> **The authors' claim that RegularGPT is able to learn long-range dependencies effectively is based on its performance on the PARITY task. However, it is not clear whether RegularGPT would be able to learn long-range dependencies as effectively on more complex tasks.**
> - We experimented with the twelve tasks in Table 1, and many of which are substantially more difficult than PARITY. RegularGPT is able to achieve perfect accuracy on them.
>
> **The authors' use of weight sharing may make RegularGPT less adaptable to different input sequences. This is because weight sharing can lead to a loss of precision, which may make it difficult for RegularGPT to learn the nuances of different input sequences.**
> - In the original submission, we thickened the number of layers ($K$ detailed in sec. 3.2 in the paper) for the task of natural language to address this problem. We also reported the effect of different $K$ in sec. 6 and Table 3.
>
> **The authors' use of adaptive depth may make RegularGPT less efficient for long input sequences. This is because adaptive depth requires the model to dynamically adjust the number of layers, which can be computationally expensive.**
> - The number of layers increases only logarithmically w.r.t the input sequence length. Concretely, chunk size of 64 can readily process sequences of length 4096 using only $log_{64} 4096=2$ layers. We consider this acceptable.
>
> **The paper does not discuss the scalability of RegularGPT. It is unclear how the model would perform on very long input sequences or on large datasets.**
> - The size of the natural language dataset used to pretrain the model is about 66GB; we believe that this is large enough to provide meaningful results. In the original submsssion, we also presented comprehensive length extrapolation results of RegularGPT on five different sequence lengths up to 8192 tokens.
>
> **The paper does not provide a detailed analysis of the training process for RegularGPT. This makes it difficult to understand how the model was trained and how it might be improved.**
> - We provide a theoretical analysis of function compositions learned by RegularGPT in sec. 7. As well, we present more detailed training setups and hyperparameters in the appendix.

---

### Meta-Review · Area_Chair_kacC · 2023-09-17

**Recommendation:** 4

**Metareview:**

Pros:
- Several of the results are strong.
- The method is simple and intuitive, a divide-and-conquer algorithm integrated into a Transformer.
- The approach is well motivated for the specific problem addressed, that of handling regular languages.
- The discussion is in depth and enjoyable to read.

Cons:
- The motivation of handling regular languages is somewhat niche.
- It is not clear why one would use a Transformer variant rather than an RNN or other existing model with a stronger inductive bias for regular languages.

---

### Decision · Program_Chairs · 2023-10-07

**Decision:**

Accept-Findings

**Comment:**

Pros:
- Several of the results are strong.
- The method is simple and intuitive, a divide-and-conquer algorithm integrated into a Transformer.
- The approach is well motivated for the specific problem addressed, that of handling regular languages.
- The discussion is in depth and enjoyable to read.

Cons:
- The motivation of handling regular languages is somewhat niche.
- It is not clear why one would use a Transformer variant rather than an RNN or other existing model with a stronger inductive bias for regular languages.